# A Confederacy of Models: a Comprehensive Evaluation of LLMs on Creative Writing

**Carlos Gómez-Rodríguez**
Universidade da Coruña, CITIC
Department of CS and IT
15071 A Coruña, Spain
carlos.gomez@udc.es

**Paul Williams**
School of Business & Creative Industries
University of the Sunshine Coast
Sunshine Coast, Australia
pwillia3@usc.edu.au

## Abstract

We evaluate a range of recent LLMs on English creative writing, a challenging and complex task that requires imagination, coherence, and style. We use a difficult, open-ended scenario chosen to avoid training data reuse: an epic narration of a single combat between Ignatius J. Reilly, the protagonist of the Pulitzer Prize-winning novel *A Confederacy of Dunces* (1980), and a pterodactyl, a prehistoric flying reptile. We ask several LLMs and humans to write such a story and conduct a human evaluation involving various criteria such as fluency, coherence, originality, humor, and style. Our results show that some state-of-the-art commercial LLMs match or slightly outperform our writers in most dimensions; whereas open-source LLMs lag behind. Humans retain an edge in creativity, while humor shows a binary divide between LLMs that can handle it comparably to humans and those that fail at it. We discuss the implications and limitations of our study and suggest directions for future research.

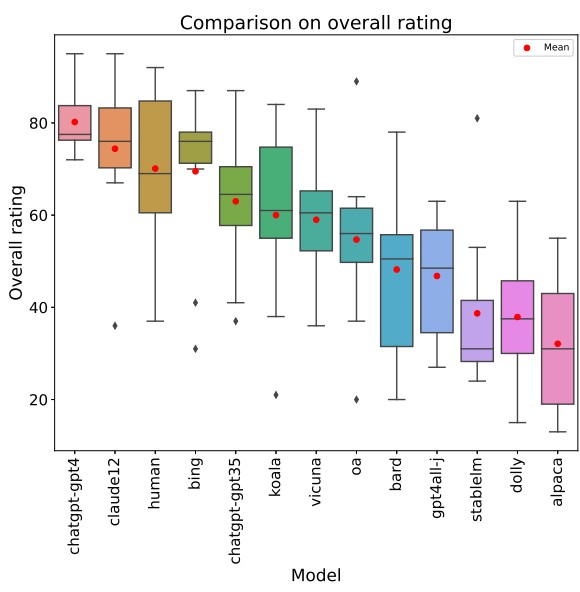

Figure 1: Box plot comparing overall ratings for stories by humans and 12 LLMs, arranged left to right by mean overall rating. Boxes show median, quartiles Q1-Q3, and whiskers at 1.5 IQR, with values outside that range plotted as outliers. Filled red circles represent means.

## 1 Introduction

In recent years, large language models (LLMs) have achieved remarkable progress in a wide range of language processing and generation tasks, such as question answering, machine translation, or text summarization, among many others (Zhao et al., 2023). This has motivated research on evaluating and comparing the performance of LLMs in various tasks, both between each other and with respect to human performance; including both task-specific evaluations (see e.g. (Jiao et al., 2023; Gilson et al., 2023)) and overarching benchmark suites that seek to provide comprehensive evaluation throughout many dimensions (Hendrycks et al., 2021; Liang et al., 2022; Srivastava et al., 2022).

Creative writing is also one application where LLMs have been observed to produce good results. According to Franceschelli and Musolesi (2023), their generated outputs in poetry or storytelling are "often of astonishing quality", and Clark et al. (2021) showed that humans cannot reliably distinguish human- from LLM-authored stories. However, and despite the amount of papers experimenting with LLMs for this purpose, an evaluation comparing the abilities of current LLMs as standalone systems for creative writing seems to be lacking.

Here, we provide such an evaluation, comparing the storytelling capability of 12 recent, instruction-aligned language models between each other and with human writers. We do so using a rubric based on established creative writing evaluation proposals (Davidow and Williams, 2016; Carey et al., 2022), but specifically adapted to the task. Our comparison is performed on a purely zero-shot setting, with a natural human prompt (based on a combat between Ignatius J. Reilly, protagonist of *A Confederacy of Dunces*, and a pterodactyl) that

has been specifically chosen to be challenging and meaningful while preventing as much as possible the option for LLMs to resort to regurgitating or adapting material from their training set.

## 2 Related work

**LLMs in creative writing**  LLMs have been used in creative writing since their first generation, with models like GPT-2 (Radford et al., 2019) or BART (Lewis et al., 2020). However, these models suffered from a lack of long-range coherence leading to contradictions or inconsistencies when generating stories (Nye et al., 2021). Thus, they were not viable as standalone story generators. Instead, they were used either with specialized fine-tuning for the task (See et al., 2019); or as components of systems that incorporated external knowledge (Guan et al., 2020, 2021), storyline planning (Tan et al., 2021), or both (Xu et al., 2020); or for co-creation with a human in the loop (Swanson et al., 2021), a line of research that has also continued with newer models (Yuan et al., 2022; Chung et al., 2022; Mirowski et al., 2023).

Here our goal is not to produce a specialized system, but to evaluate the performance of LLMs by themselves as creative writers. Thus, we focus on the purely zero-shot setting, where a generalistic LLM is asked to write a story with no extra fine-tuning, in-context learning (Dong et al., 2023), prompt engineering or additional components. This has only become viable with the extra coherence and consistency in long texts provided by newer LLMs, especially those that are aligned to follow instructions with instruction tuning (Wei et al., 2022; Sanh et al., 2022) or reinforcement learning with human feedback (Ouyang et al., 2022).

To our knowledge, there was no previous work in this line. In fact, evaluation in creative writing is a conspicuous gap in LLM evaluation benchmarks: the huge BIG-bench suite (Srivastava et al., 2022) currently has over 200 tasks, but does not include any creative writing, and HELM (Liang et al., 2022) cites it as an "aspirational scenario" for future work. This likely owes to benchmarks focusing on easily-automatable metrics, whereas the gold standard for creative writing is human evaluation (Belz and Reiter, 2006), which is much costlier.

The closest previous work to our proposal is the recent preprint by Xie et al. (2023), where GPT-3 is compared to previous storytelling systems via human evaluation. However, there are several impor-

tant differences with respect to our work: (1) they use prompt-based learning, providing examples to adapt the model to the task, rather than a purely zero-shot conversational prompt, (2) they evaluate a single LLM while our goal is to compare LLMs, and (3) they use pre-existing story datasets, which increases the risk of models benefitting from similar stories present in their training set, something that we have tried to avoid as described below.

In another recent preprint, Garrido-Merchán et al. (2023) generate Lovecraftian horror literature. However, they also focus on a single LLM (GPT-4), using careful prompt engineering to optimize its performance rather than a pure zero-shot setting, and evaluation is only on whether humans can distinguish AI-generated from real stories (concluding that, in those circumstances, they cannot). Sawicki et al. (2023) apply a similar evaluation (but automated) to Whitmanian poems generated by three versions of GPT, also with a negative result.

Finally, concurrently with our study, a preprint by Chakrabarty et al. (2023), released a few months after our submission, evaluates three LLMs for creative writing in a more similar way to ours: they apply human evaluation to compare stories by humans and LLMs in a zero-shot setting. However, there are important differences in methodology and scope between both studies. A comprehensive comparison will be made in Section 5, following the exposition of our methods and results.

**Creative writing evaluation**  Creative Writing is a challenging and complex performative language act that requires a number of skills, such as an expertise in craft, cultural and literary competency, linguistic fluency, coherence, complex connotative and metaphorical levels of understanding, innovation, originality and imagination, to name a few.

The craft of writing involves innovation with style and voice, needs a fundamental understanding and use of structural elements (grammar, spelling, punctuation), craft elements (plot, character, setting, point of view and imaginative capacity, such skills defined by Bloom as 'putting elements together to form a coherent or functional whole; reorganizing elements into a new pattern or structure through generating, planning, or producing' (Anderson and Krathwohl, 2001, p.21). Evaluation of creative writing therefore must take into account all these factors, and assessment in university Creative Writing courses is usually based on a rubric that attempts to measure the basic elements of narrative

craft, as well as the specific requirements on the assignment (Kroll, 1997; Norris, 2013; Davidow and Williams, 2016; Wise and van Luyn, 2020; Carey et al., 2022).

## 3 Materials and Methods

### 3.1 Task

The chosen task to compare the LLMs under consideration is defined by the following prompt:

> Write an epic narration of a single combat between Ignatius J. Reilly and a pterodactyl, in the style of John Kennedy Toole.

The prompt is provided to the models from a fresh state, without previous context.

We believe this task is particularly adequate to challenge the capabilities of models for creative writing, for the following reasons:

- It is a non-standard, "wacky" scenario that has been invented for the occasion, so it is very unlikely that the systems' training sets contain coincident or similar tasks, or pieces of stories that can be reused for the task. No information about this task was posted to the Internet or disseminated in any other way before the LLMs were prompted.

- It features a specific literary character, Ignatius J. Reilly, so we can evaluate the models on how they capture the personality of the character. At the same time, this character appeared in only one book, and does not seem to have been the target of fan fiction. This makes the task more challenging due to having to capture the personality of the protagonist from scarce material, while making it unlikely that the model can just reuse material from existing stories.

- In turn, *A Confederacy of Dunces* is the only work of its author John Kennedy Toole, so the author's style also needs to be captured from scarce material.

- This novel is widely considered to be a classic of comic fiction, and won the 1981 Pulitzer Prize in the Fiction category. Thus, writing a story about its protagonist in the author's style sets an adequately high bar.

- The genre requires humor, which is considered to be an especially subtle feature of human language and challenging for machines, including LLMs, to exhibit (Jentzsch and Kersting, 2023).

- While the task is challenging due to putting together two unlikely antagonists, the prompt's level of detail is open-ended enough to give ample space for creativity, as no specifications are made about setting, weapons, outcome or other aspects of the story.

### 3.2 Models

We gave the task to a confederacy of large language models, composed of all such models we could find that (1) were available to the authors by April 20 2023, which was the cutoff date to build our corpus of stories, and (2) were adjusted to conversational settings and instruction-following by using techniques like instruction tuning (Wei et al., 2022; Sanh et al., 2022) or reinforcement learning with human feedback (Ouyang et al., 2022). This is in contrast to "vanilla" language models configured to just predict the next word, like plain GPT-3 (Brown et al., 2020) or Llama (Touvron et al., 2023), which generally cannot handle natural prompts like the one we use. We only included distinct models, not front-ends to the same model (but we did include derived models with substantial additions, like Bing Chat which is claimed to use GPT-4 but adds search capabilities, or various models that were fine-tuned from Llama weights). For models that came in a variety of parameter sizes, we used the largest one, or the largest we could execute with local or remote resources. For models with several available versions, we used the latest available, except in the case of ChatGPT where we included both the GPT-3.5 and GPT-4 versions, due to the wider availability of 3.5 (the latest version offered for free at cutoff time) and the lack of information on whether GPT-4 is an incremental improvement or a different model with its own tradeoffs.

This selection yielded the following 12 language models. We list them in alphabetical order as chronological ordering would be challenging, due to closed releases, opaque updates from some of the commercial products, and many of the models being released almost simultaneously:

**Alpaca** (Taori et al., 2023), a Stanford model fine-tuned from Llama (Touvron et al., 2023) on instruction data generated with the self-instruct

methods of (Wang et al., 2022). We use the 13B-parameter version, the largest available at cutoff.

**Bard**, Google's experimental conversational LLM offering, claimed to be based on a lightweight version of LaMDA (Thoppilan et al., 2022). It can use content from the web to answer questions. Model details have not been made public.

**Bing Chat**, an LLM offered by Microsoft's Bing search engine. Claimed to use GPT-4[1], further technical details have not been made public. The model performs web searches and uses the results to augment its context window with relevant information. It can also provide links to sources for its claims (although this is not relevant for our creative writing task, where no such links were provided or needed). We used its Creative mode, the obvious fit for our task. A problem worth mentioning is that we found the model to be subject to heavy censorship, which affected our experiment: in most prompting attempts, the story would be deleted by the filtering system before being finished. When this happened, we just reset and re-prompted the model, repeating the process until a full story was obtained. Over 100 tries were needed to obtain 5 non-censored stories. We are aware that this may introduce bias (as non-censored stories may have a different quality distribution than what the model could potentially generate without the filter) but this is unavoidable from our end, since we cannot bypass moderation. In any case, the sample does reflect what a user can obtain from the end product, as the censored stories are out of reach.

**ChatGPT with GPT-3.5**, an OpenAI successor to the 175B-parameter GPT-3 model (Brown et al., 2020) which was tuned using reinforcement learning with human feedback, namely a variant of the InstructGPT method by Ouyang et al. (2022). We used the March 23 version provided by OpenAI's free ChatGPT service.

**ChatGPT with GPT-4**, the most advanced language model released by OpenAI at cutoff time. A description of the model is available in (OpenAI, 2023), although essential technical details like the number of parameters have not been published. We used the March 23 version provided by OpenAI's ChatGPT Plus service.

**Claude** is a language model trained by Anthropic. While details about its implementation are not public, it is known to be a sucessor of the model

described in (Bai et al., 2022), a 52B-parameter model aligned to be helpful with Constitutional AI, a list of guiding principles provided to the model, combined with a mix of supervised learning and reinforcement learning with AI feedback. We used version 1.2 of the model.

**Dolly 2.0** (dolly-v2-12b), a 12B-parameter language model trained by Databricks, derived from EleutherAI's Pythia-12B model (Biderman et al., 2023) after fine-tuning on a 15K instruction corpus. At cutoff date, it was the only available conversational LLM where all of its components could be considered fully open source[2], as the code, weights and instruction datasets all have open-source licenses compatible with any use, including commercial use, and no data from proprietary systems like ChatGPT has been used for finetuning.

**GPT4All-J** (Anand et al., 2023b), an improvement over its predecessor GPT4All (Anand et al., 2023a). The base model is the 6B-parameter GPT-J (Wang and Komatsuzaki, 2021), which has been fine-tuned on a dataset expanded from a mix of existing sources.

**Koala** (Geng et al., 2023), a model fine-tuned from Llama (Touvron et al., 2023) by researchers from the university of Berkeley, on a variety of dialogue data obtained from the web. We use the 13B-parameter version.

**OpenAssistant** (Köpf et al., 2023) is an LLM fine-tuned on a large, free, human-generated conversation corpus created by a crowdfunding effort involving over 13,500 volunteers. We used the OA-SFT-Llama-30B model, fine-tuned from the 30B-parameter Llama (Touvron et al., 2023) model.

**StableLM** is Stability AI's series of language models. We used StableLM-Tuned-Alpha-7B. With 7B parameters, this is the largest model available (at cutoff time) among a series of models trained on a dataset built from The Pile (Gao et al., 2021) and fine-tuned on a combination of conversational LLM corpora.

**Vicuna** (Chiang et al., 2023) is another member of the family of models obtained by fine-tuning Llama (Touvron et al., 2023), in this case with user-shared conversations with ChatGPT. We used the 13B-parameter version of the model.

### 3.3 Evaluation rubric

The creative writing rubric was designed for assessment of creative writing assignments in uni-

---

| ID | Description |
|---|---|
| 1 | Overall/holistic/cohesive readability of the story (not just a compilation of elements). |
| 2 | Use of key narrative elements - vocabulary choice, imagery, setting, themes, dialogue, characterisation, point of view. |
| 3 | Structural elements and presentation which reflects the control of structural elements such as spelling, grammar, punctuation, paragraphing, and formatting. |
| 4 | Overall plot logic: hook, conflict, initial crisis, rising and falling action, denouement/ resolution (Freitag's pyramid). |
| 5 | Creativity/innovation/originality/ research—credibility, new knowledge, avoidance of cliché and derivative tropes. |
| 6 | Incorporation of the John Kennedy Toole style of writing using the indicators/ characteristics listed. |
| 7 | Understanding and habitation of the epic genre of heroic/legendary adventure. |
| 8 | Description and credibility of a single combat scene. |
| 9 | Accurate inclusion of two main characters Ignatius J. Reilly and a pterodactyl in action and description. |
| 10 | Use of a characteristically dark humorous tone. |

[1] https://blogs.bing.com/search/march_2023/Confirmed-the-new-Bing-runs-on-OpenAIâĂŹs-GPT-4

Table 1: Creative writing evaluation rubric. All items are scored out of ten points. Marking guideline: Emerging 1-4, Competent 5-8, Sophisticated 9-10.

versity creative writing courses, and is taken in part from a university textbook by one of the authors of this article, *Playing with Words* (Davidow and Williams, 2016) and an article that justifies the use of this rubric (Carey et al., 2022). This rubric evaluates creative production in five holistic craft-based criteria and measures craft skills based on a writing style outlined in the article: among others, Flaubert's insistence on *le mot juste* (the right word or expression), Strunk and White's *The Elements of Style* (2008[1918]), George Orwell's rules for concreteness and clarity (Orwell, 1946); and Annie Dillard's rules for writing good prose (Dillard, 1981).

The rubric for this AI task adds five more criteria which address the specific prompt requirements, such as genre, style, tone, character and action. Each of the ten criteria is awarded 10 points out of a total 100 points. The rubric has been specifically designed to measure the quality of writing craft, to avoid formulaic, rule-based writing and to address the very specific task addressed here.

The criteria are detailed in Table 1, with more details given in the Appendix C. The holistic scale (emerging, competent, sophisticated) guides human raters to assess holistically: 'a holistic scale measures the relative success of a text but does so through a rubric that incorporates many of the traits in analytic scoring as heuristics towards a conception of a whole rather than as a sum of autonomous components' (Perelman, 2018, p.16).

### 3.4 Evaluation methodology

We prompted each of the LLMs 5 times with the prompt given in Section 3.1. Each prompt was made from a fresh state, i.e., in a zero-shot setting without any previous context that could help guide the models. The resulting stories had an average of

379 words (std = 248, min = 23, max = 1223).

Then, we also asked 5 human writers to each write a story following the same prompt. For uniformity, we suggested a length range coherent with the LLM-generated stories (250 to 1200 words). The writers were Honours and postgraduate Creative Writing students that volunteered for the task, and all of them studied the specific task requirements (e.g. John Kennedy Toole's style) before writing their stories. However, they were not given access to the AI-generated stories and they were instructed not to use LLMs at all to help them write.

The result is, thus, a corpus of 60 AI-generated stories (5 for each of the 12 considered LLMs) plus an additional 5 human-generated stories, all in plain text format. The corpus is available at https://doi.org/10.5281/zenodo.8435671.

The only preprocessing made to the stories is that (1) we removed leading sentences that described the task, often present in LLM answers (e.g.: "Here is a potential epic narration in the exaggerated style of John Kennedy Toole's A Confederacy of Dunces:") (2) we removed titles from stories that had them, and (3) we unified paragraph formatting, leaving one line between paragraphs in all the plain text files. Other than these changes, made for uniformity and to preserve the blindness of the rating process, we left the text as it was.

We recruited 10 raters, also Honours and postgraduate Creative Writing students that were acquainted with the specific requirements of the task, and we instructed them to grade stories according to the rubric. Since the raters were volunteers, to keep the workload low, each rater did not rate all the stories. Instead, we divided the 65 stories into 5 groups of 13 stories each (each group containing one story by each LLM, plus one story by a human) and assigned one rater to each group. In this way,

| Rubric item | 1 | 2 | 3 | 4 | 5 | 6 | 7 | 8 | 9 | 10 | overall |
|---|---|---|---|---|---|---|---|---|---|---|---|
| chatgpt-gpt4 | **8.7**±0.8 | **8.7**±0.7 | **8.4**±1.3 | **8.3**±0.7 | 7.6±1 | **8.0**±1.2 | **8.1**±1.4 | **8.5**±0.8 | **7.9**±1.6 | 6.0±2.8 | **80.2**±7.3 |
| claude12 | 8.0±1.7 | 8.0±1.6 | 8.1±1.2 | 7.9±1.8 | 7.1±2.3 | 7.5±2 | 6.4±2.2 | 7.5±1.8 | 7.4±2.5 | **6.5**±2.5 | 74.4±15.9 |
| human | 7.3±2.3 | 7.8±1.8 | 7.3±1.7 | 7.2±1.8 | **8.0**±2 | 7.2±2.4 | 4.9±2.1 | 6.3±2.2 | 7.7±2.1 | 6.4±3.4 | 70.1±17.4 |
| bing | 7.8±2 | 7.5±2.2 | 7.9±1.7 | 7.4±2.1 | 7.0±1.6 | 6.8±2.4 | 5.3±2.9 | 6.2±2.1 | 7.4±2.2 | 6.2±2.6 | 69.5±18.4 |
| chatgpt-gpt35 | 7.5±2 | 6.5±2.4 | 8.1±1.3 | 7.0±2.2 | 5.4±2.5 | 5.3±2.4 | 6.8±1.5 | 7.6±1.2 | 5.5±2.5 | 3.3±2.8 | 63.0±15.4 |
| koala | 7.5±2.5 | 6.7±2.2 | 8.2±1.2 | 6.8±2.6 | 5.8±2.3 | 4.8±2.7 | 5.8±2.4 | 5.5±1.8 | 5.5±2.3 | 3.4±3.2 | 60.0±19.2 |
| vicuna | 7.9±1.7 | 6.7±1.6 | 8.1±1.3 | 7.0±1.6 | 5.1±1.9 | 4.6±2.3 | 5.7±2.3 | 6.1±1.9 | 5.4±2.7 | 2.4±1.9 | 59.0±13.8 |
| oa | 7.2±2.2 | 5.8±2.4 | 7.2±2.5 | 6.2±2.6 | 4.9±2.1 | 3.9±2.4 | 5.8±2.4 | 6.5±2.2 | 4.3±2.3 | 2.9±3.1 | 54.7±18 |
| bard | 6.5±2.5 | 4.9±2.1 | 6.8±1.9 | 5.5±2.7 | 3.9±2.1 | 3.8±2.5 | 4.7±2.6 | 4.6±2.7 | 5.0±2.4 | 2.5±2 | 48.2±20.1 |
| gpt4all | 6.5±2.2 | 5.4±1.7 | 7.2±1.7 | 6.5±2.1 | 4.1±2.2 | 2.4±2.2 | 5.4±2.5 | 5.6±2.4 | 2.5±1.4 | 1.2±0.8 | 46.8±13.1 |
| stablelm | 5.5±1.8 | 5.0±2.5 | 6.6±1.9 | 3.8±2 | 3.2±1.5 | 2.1±2.2 | 4.4±1.9 | 3.8±2 | 2.9±2.6 | 1.4±1.5 | 38.7±17.2 |
| dolly | 4.6±2.2 | 5.0±2.2 | 5.6±2.5 | 3.2±1.9 | 4.2±2.8 | 3.1±2.2 | 4.4±1.9 | 3.3±1.8 | 3.0±2 | 1.5±1.5 | 37.9±13.6 |
| alpaca | 5.2±3.1 | 3.1±1.4 | 4.9±3 | 4.2±1.9 | 1.9±1 | 2.0±1.4 | 3.7±3 | 3.9±2.8 | 2.1±1.5 | 1.1±0.6 | 32.1±15.7 |
| average | 6.9±2.1 | 6.2±1.9 | 7.3±1.8 | 6.2±2 | 5.2±2 | 4.7±2.2 | 5.5±2.3 | 5.8±2 | 5.1±2.2 | 3.4±2.2 | 56.6±15.8 |

Table 2: Results for each rubric item, as well as overall score. Each cell shows average ± standard deviation for the ratings achieved by a given model (or human writers) on a given rubric item. The bottom line shows the average among all models (and human writers). Models are sorted by overall score. The best result for each rubric item is highlighted in boldface.

we ensure (1) that we have at least two ratings per story, allowing us to measure inter-rater agreement, (2) that comparisons are fair, in the sense that no LLM (or the humans) is advantaged by being assigned more lenient raters, because each LLM (and humans) receives exactly one rating by each of the 10 raters, and (3) since each rater always gets one story from each model (and one human), we can expect that each will be rating a diverse set of stories covering a wide range of ability levels, which helps the marking process as it allows for comparative analysis between various performances, enabling more accurate pinpointing of each story's quality.

Stories were assigned random identifiers before sending them to raters, so that the process was blind: to avoid biases, raters knew that they would be evaluating human and AI-generated stories, but were unaware of the origin of each story.

Raters were sent all stories at once and they were free to go back and change the ratings of previously-rated stories. In addition, all of them were experienced assessors in terms of Creative Writing texts, with previous experience in applying the scale. These precautions mitigate the need for specific calibration (Karpinska et al., 2021) that would strain our resources.

# 4 Results

## 4.1 Agreement

To gauge the reliability of our results, we compute inter-rater agreement between the two ratings given to each story for each individual rubric item. We use linearly weighted Cohen's kappa (Cohen, 1968), which is appropriate for ordinal scales like ours, obtaining a value of $0.48$, 95% CI $[0.43, 0.54]$. This is interpreted as "moderate agreement", which is a positive result taking into account the obvious subjectivity involved in rating stories. If we instead focus on overall scores (sums of rubric items), the Pearson correlation between the scores given to each story by each group of raters is $0.58$ ($p < 0.00001$), again indicating a reasonable degree of consistency between raters given the subjectivity of the task.

## 4.2 General overview

Table 2 shows a comprehensive overview of the ratings that each of the LLMs (and humans) obtained for each rubric item, as well as in terms of overall score. Additionally, a box-and-whisker plot comparing overall score can be seen in Figure 1.

ChatGPT with GPT-4 generates the best-rated stories, both in terms of overall score and in 8 out of 10 of the individual rubric categories. However, human writers are rated best in terms of originality (rubric item 5), and Claude was rated best in the use of dark humor (rubric item 10), with humans a close second. GPT-4 is also remarkably consistent, showing low standard deviations not only with respect to human writers (which is expected, as our human stories were authored by five different humans, whose skill levels may vary) but also with respect to the rest of the LLMs.

If we compare LLMs to each other, the best performances correspond to commercial offerings, including (apart from the aforementioned GPT-4) Claude, Bing Chat and the GPT-3.5 version of ChatGPT. Open-source models are clearly behind, with the best (Koala) achieving $60.0$ overall score, contrasting with the $80.2$ obtained by GPT-4. Although the best-performing LLMs are generally better across the board, some idiosyncrasies can be observed: e.g., GPT-4 tops almost all rubric items

but is outperformed by two LLMs at humor.

When we compare LLMs to human writers, significance testing on overall score (2-tailed t-test assuming unequal variances) fails to detect significant differences between humans and the top 6 AI models with $\alpha = 0.05$. Only the 6 bottom AI models are significantly worse than humans at this significance level. Note, however, that the test has a low statistical power due to the small sample size (10 ratings per model). If we instead perform a test on individual metrics, so our sample size is 100 (with the null hypothesis being no difference between humans and each LLM in random individual metric scores), then GPT-4 is identified as significantly better than the human writers ($p = 0.00031$), Claude and Bing's scores are not significantly different from those of humans, and all the rest of the LLMs score significantly worse than humans.

Looking at individual metric scores, structural elements (rubric item 3) are the easiest category (with an average rating across all stories of 7.3, and all models but one obtaining at least a 5 on average). Humor (rubric item 10) is clearly the hardest, with an average score of 3.4, and we will analyze it in more detail below. Incorporating John Kennedy Toole's style is the second hardest, with 4.7. Comparing humans to LLMs, humans (as already mentioned) excel at originality and humor, but are clearly behind the best LLMs in terms of readability (item 1), where they are outperformed by 6 LLMs, and even more so in use of the epic genre (item 7), where they score 4.9 and are outperformed by 8 LLMs.

We now analyze in more detail some of the individual items that show more interesting comparisons between human writers and LLMs.

### 4.3 Humor

Figure 2 shows a box plot that complements the information on Table 2 for the humor rubric item. The results for this item have two interesting characteristics. Firstly, it is clearly the most difficult rubric item, with an average score across models of 3.4, and the best obtaining 6.5. Even humans obtain a lower score in humor than in most items, which may be a consequence of humor being highly subjective. Secondly, as evidenced both in the table and plot, there is a rather stark binary divide between the contenders that "get" humor and those that do not: Claude, Bing and GPT-4, together with the human writers, obtain average scores between

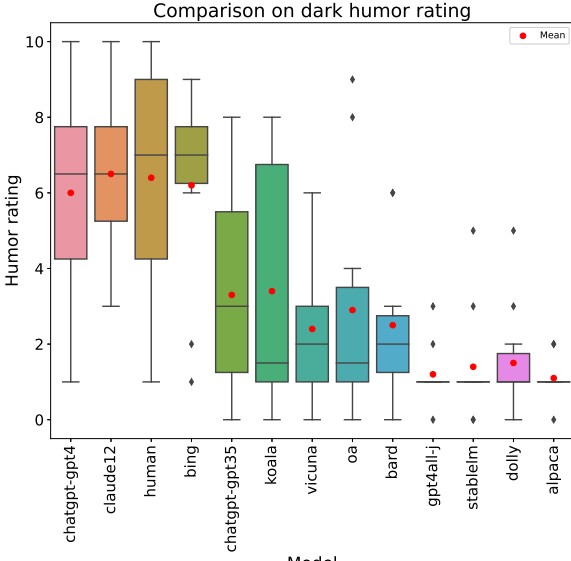

Figure 2: Box plot comparing humor ratings for stories generated by humans and 12 LLMs, sorted left to right by mean overall rating. Notation as in Figure 1.

6 and 6.5; whereas the rest of the models achieve very low scores of 3.4 or less. Significance testing also confirms this divide: despite the small sample size of 10 humor ratings per model, a 2-tailed t-test with $\alpha = 0.05$ confirms that the models in the second group are significantly worse than the human writers, as well as the LLMs in the first group. This suggests that grasping human humor might be an emergent ability of larger LLMs.

In this respect, a recent preprint (Jentzsch and Kersting, 2023) concluded that ChatGPT has "a limited reflection of humor" and "cannot yet confidently create intentionally funny original content". This study used the GPT 3.5 version of ChatGPT, so it is in line with our results (in which that model obtains an average humor score of 3.3). However, as we have seen, more powerful LLMs have overcome that limitation, as their generated stories are clearly rated as humorous.

### 4.4 Creativity

We now focus on rubric item 5, which rates creativity and originality, as it is a hallmark of creative writing and also the only category where human writers have outperformed all the LLMs in our analysis. Figure 3 shows a box plot that complements the information on Table 2.

The same three LLMs that stood out in the humor category are also the best in terms of creativity, although the difference is not as stark. Regardless, a t-test still distinguishes both groups as it shows all

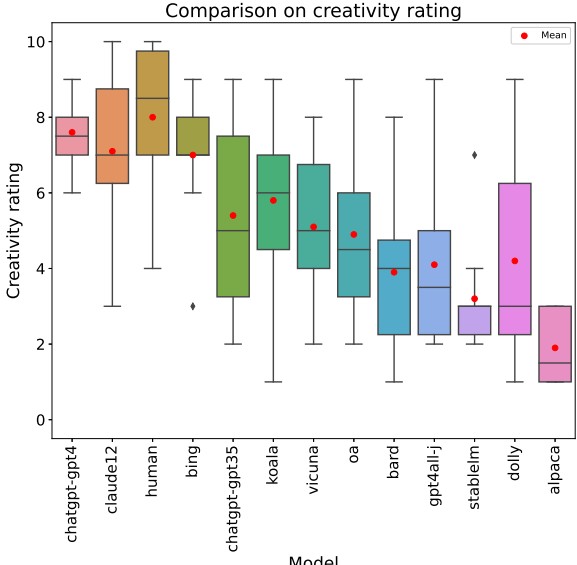

Figure 3: Box plot comparing creativity ratings for stories generated by humans and 12 LLMs, sorted left to right by mean overall rating. Notation as in Figure 1.

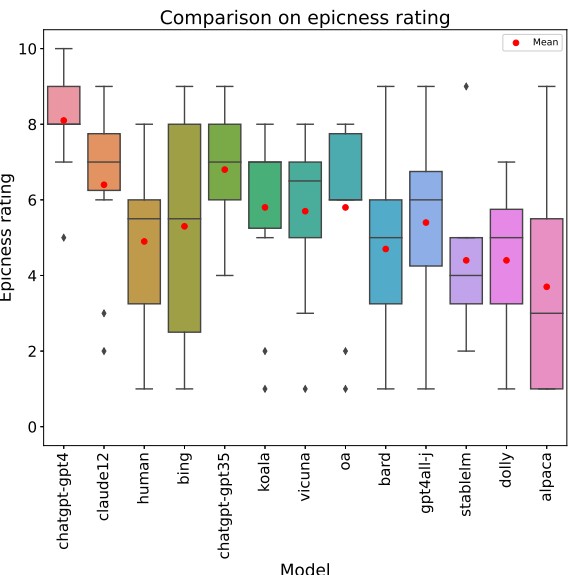

Figure 4: Box plot comparing epicness ratings for stories generated by humans and 12 LLMs, sorted left to right by mean overall rating. Notation as in Figure 1.

the rest of the LLMs to be rated as significantly less creative than our human writers, while for these three we cannot reject the null hypothesis that they are as original as the human writers.

Overall, from our results and in terms of human perception of the output, the answer to whether LLMs can produce creative stories (Franceschelli and Musolesi, 2023) is yes, although humans still retain an edge in this respect.

### 4.5 Epicness

Finally, we analyze rubric item 7 (understanding and habitation of the epic genre) for the opposite reason as in the previous section: it is the item where humans do worst compared to LLMs (see Table 2). A box plot is provided in Figure 4.

In this case, the results have a more atypical profile, with substantial difference with respect to overall scores. Two models perform significantly better than the human writers ($\alpha = 0.05$): both versions of ChatGPT. Other six models obtain better average rating than humans, but the difference is not detected as significant.

Interestingly, Bing clearly lags behind both Chat-GPT versions, despite being based in GPT-4. This might be related to bias introduced by the system's censorship. On the other hand, some models whose overall scores are in the bottom half (OpenAssistant, GPT4All) are reasonably good at epic narration, outperforming humans and Bing (which are better than them in almost all categories).

## 5 Discussion

We have evaluated recent LLMs on a creative writing task in English, using a carefully-designed scenario to provide a demanding challenge and avoid confounding factors like training data memorization (Carlini et al., 2023). To our knowledge, this is the most thorough evaluation of LLMs on creative writing conducted so far, both in terms of scope (12 LLMs considered, plus comparison to human writers) and detail (using human evaluation with a 10-item rubric based on established creative writing evaluation practices).

Simultaneously to our work, the recent preprint by Chakrabarty et al. (2023) provides an evaluation of three of the top-performing commercial LLMs (ChatGPT, GPT-4 and Claude) for creative writing. This approach is close to ours, as it uses the models in a zero-shot setting and evaluation is performed by humans using a specific rubric. However, there are important methodological differences between both studies, which we summarize here:

1. The human stories used by Chakrabarty et al. (2023) are stories published in the New Yorker, by highly successful authors (including Nobel prize winners), whereas ours are written by Creative Writing students.

2. In their setting, the human-written stories are pre-existing (and selected for publication in the New Yorker, as mentioned above) so their

writers were unconstrained when they created them, while the LLMs have to adapt to write an alternative story with the same plot. In ours, humans and LLMs are given the exact same prompt to work with.

3. In terms of length, the stories they work with are over thrice larger than ours on average. In addition, while both studies try to make sentence lengths similar between humans and LLMs, in their case the human writers originally wrote their stories unconstrained (or under loose constraints) and the LLM-generated stories were calibrated to have similar lengths by an iterative prompting process. In our case, the LLMs were unconstrained in terms of length, and the human writers were suggested to target a length range loosely similar to LLM-generated stories. Thus, with respect to theirs, our approach has the disadvantage of a looser control on story length, but the advantage of using a single zero-shot prompt.

4. Their study spans a variety of story prompts, while we focus on a single prompt and setting. The flip side is that our rubric can be adapted to specific requirements like humor and Toole style, whereas theirs is necessarily more generic. In addition, our narrower focus allows us to have LLMs generate several alternative stories, so we can perform more statistical analysis: we consider the distribution within each LLM and perform statistical testing, which cannot be done in Chakrabarty et al. (2023)'s setting as they generate a single story per prompt and LLM.

5. Since their study is based on existing stories that are published online, there is the possibility that some are contained in the tested LLMs' training data. In our case, we designed the study to prevent training data reuse.

6. The rubrics are different: Chakrabarty et al. (2023) use a rubric based on the Torrance tests of creative thinking (Torrance, 1974).

The outcome of this study is substantially different from ours, with LLM-generated stories rated clearly behind human-authored ones. This is not surprising considering the methodological differences: in particular, differences 1 and 2 in the list above clearly set a higher bar for LLMs, as they are compared to highly successful human stories by top authors that wrote freely and the LLMs are asked to adapt to their plots. We hypothesize that these are the main reasons for the difference in outcome. On the other hand, item 5 in the list above could in principle benefit LLMs, and there are other factors that could benefit humans or LLMs in non-obvious ways (including items 3, 4 and 6, as well as different story genres and target lengths). This underscores the need of more studies in this area.

# 6 Conclusion

The results show that state-of-the-art LLMs can perform a creative writing task at a very competent level, with the top two (ChatGPT with GPT-4 and Claude) achieving high scores that outperform human writers in most rubric categories. While we must be careful not to take this as evidence of "superhuman storytelling" (both because our sample size is not enough to draw such categorical conclusions, and because our 5 human writers are not necessarily representative of human writing ability as a whole); it does at least strongly suggest that these models' stories are not distinguishably worse than those by reasonably-trained humans. This is even more remarkable given that we did not use any in-context learning or other techniques to optimize the LLMs for the task, but just a straightforward prompt from a fresh state, so it is possible that even better results are achievable with careful prompting.

Our analysis also shows that the best results are achieved by commercial LLMs, with open-source models clearly lagging behind at the moment.

Looking at individual characteristics, humans retain the lead in originality, while LLMs tend to excel in more technical aspects like readability or structure. Humor is an especially challenging aspects where most LLMs utterly fail, but the best three models do succeed at achieving human-like ratings, contrasting with results on older LLMs that showed their lack of grasp of human humor (Jentzsch and Kersting, 2023).

Interesting avenues for future work include evaluation of different literary genres, languages other than English, and studying whether the quality of the generated stories can be improved with prompt engineering or fine-tuning.

Selected stories from our corpus (available at https://doi.org/10.5281/zenodo.8435671, together with all rating data) are in Appendix E.

## Limitations

**Commercial LLMs and reproducibility**  While some of the LLMs considered are proper scientific artifacts, trained with a documented methodology and whose code and weights are available, others are closed commercial products and there is little public information about them, hindering reproducibility. While we have reported version numbers (where available) and access dates are provided in Appendix A, apart from publishing the generated outputs so that the rating process is reproducible, the prompting/generation process may not be reproducible in the future for these models as some of these products are updated without notice, and without providing access to previous versions. However, we believe that including commercial models is valuable, as they are widely considered to provide the best quality results at the time of writing (which has been confirmed by our analysis), and these data points can still be used as a measuring stick against which to compare open models in the present and future.

**Limitations of the analysis**  Rating creative writing is necessarily a highly subjective process. Furthermore, since our raters were volunteers, we did not ask each of them to mark the full 65 stories in the corpus but just a subset, so our sample size is limited. We have provided the necessary details so that the reader can assess the variability of the data (sample sizes, standard deviations, and inter-rater agreement, which is reasonably high given the subjectivity of the task); and we have been careful not to make overarching claims. In this respect, we have also taken into account that our sample of human writers cannot be assumed to be representative of "human creative writing ability" as a whole, but is only provided as a reference point of interest; and that our evaluation is focused on a specific genre, so claims of the form "LLMs are better/equal/worse than humans at creative writing" cannot be made with an evaluation like ours.

**Scope**  Our analysis focuses on a specific genre, and on English language, so the results do not necessarily generalize to other genres and/or languages. However, conducting a wider evaluation in this respect would not be possible with our resources, so we chose to fix these variables and focus on conducting a detailed evaluation on a large number of LLMs instead.

## Ethics Statement

While the use of conversational LLMs has raised various ethical challenges, creative writing has been argued to be one of the best uses for these tools from a human-centered AI point of view, as long as AI-generated stories are identified as such to avoid misleading readers or publishers (Sison et al., 2023). In our study, raters were blinded to story authorship but they were previously informed that they would be dealing with AI and human-generated stories. In the published corpus, each story is identified as human or AI-authored.

All participants in the evaluation (as raters or writers) were volunteers, and the demand on their time was kept accordingly low.

## Acknowledgments

The first author was funded by the European Research Council (ERC), under the Horizon Europe research and innovation programme (SALSA, grant agreement No 101100615), ERDF/MICINN-AEI (SCANNER-UDC, PID2020-113230RB-C21), Xunta de Galicia (ED431C 2020/11), and Centro de Investigación de Galicia "CITIC", funded by the Xunta de Galicia through the collaboration agreement between the Consellería de Cultura, Educación, Formación Profesional e Universidades and the Galician universities for the reinforcement of the research centres of the Galician University System (CIGUS).

We thank Olga Zamaraeva for comments on preliminary versions of this work, and two anonymous reviewers for their helpful comments. Last, but not least, we thank our volunteers who participated in the writing and grading of stories, in alphabetical order: Jayda Franks, Bree Glasbergen, Ola Kwintowski, Jay Ludowyke, Kyle Mackenzie, Kirsty Maclachlan, Caitlin Noakes, Rachelle Raco, Kylie Ryan and Josephine Stewart. Credit for each individual story can be found in the corpus.

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

## A  Model access dates

Table 3 shows the date in which the stories were generated for each of the models. For future experimental reference, we highlight that the initial public disclosure of this paper online occurred on 2023-10-09. Before this date, only the human authors and raters were aware of the project from May 2023, and anonymous reviewers had access from June 23, 2023. Consequently, LLMs with a knowledge cutoff prior to 2023-10-09 are likely to have no or minimal risk of training set contamination.

## B  Hyperparameters

We did not tweak any hyperparameters of the models. In the case of commercial models, we just ran the model as it is presented in their respective web user interfaces, except in the case of Bing Chat where we chose Creative mode. For open-source models, we used the default parameters from the web UI provided at `https://chat.lmsys.org/`, which set temperature to 0.7.

| Model | Access date |
|---|---|
| alpaca | 2023-04-07 |
| bard | 2023-04-11 |
| bing | 2023-04-11 |
| chatgpt-gpt35 | 2023-04-11 |
| chatgpt-gpt4 | 2023-04-14 |
| claude12 | 2023-04-04 |
| dolly | 2023-04-14 |
| gpt4all-j | 2023-04-14 |
| koala | 2023-04-07 |
| oa | 2023-04-16 |
| stablelm | 2023-04-20 |
| vicuna | 2023-04-07 |
| humans | 2023-05-01 to 2023-05-12 |

Table 3: Access dates for each model (and dates of writing for the human stories), in YYYY-MM-DD format.

## C  Detailed rubric information

The creative writing rubric was designed for assessment of creative writing scripts in university creative writing courses in order to evaluate these above competencies, criteria 1-5 to measure general creative writing capacities, and criteria 6-10 to measure specific task related proficiency. Each of the ten criteria is awarded 10 points out of a total 100 points. The rubric has been specifically designed to measure the quality of writing craft and to avoid formulaic, rule-based writing.

1. Overall/ holistic/ cohesive readability of the story (not just a compilation of elements).

2. Use of key narrative elements - vocabulary choice, imagery, setting, themes, dialogue, characterisation, point of view.

3. Structural elements and presentation which reflects the control of structural elements such as spelling, grammar, punctuation, paragraphing, and formatting

4. Overall plot logic: hook, conflict, initial crisis, rising and falling action, denouement/ resolution (Freitag's pyramid)

5. Creativity/innovation/originality/ research—credibility, new knowledge, avoidance of cliché and derivative tropes

6. Incorporation of the John Kennedy Toole style of writing using the indicators/ characteristics listed below

7. Understanding and habitation of the epic genre of heroic/legendary adventure

8. Description and credibility of a single combat scene

9. Accurate inclusion of two main characters Ignatius J. Reilly and a pterodactyl in action and description (see below for character description)

10. Use of a characteristically dark humorous tone.

The 1-10 scale is divided into three ranges:

- Emerging (1-4): stories in this range demonstrate an early grasp of storytelling elements, but falter in execution or depth. When evaluating humans, they correspond to novice writers who need feedback and guidance to improve the story.

- Competent (5-8): stories that showcase a good grasp of the storytelling principle being evaluated (coherent plot, well-defined characters, etc.). While there might be room for improvement, these stories effectively engage the reader and convey their intended messages.

- Sophisticated (9-10): these stories exhibit exceptional mastery of the aspect being evaluated, resulting in a compelling and memorable read.

**Toole style** We provided raters with detailed information about the plot, setting, imagery, tone, characters, main protagonist, and derivative/imitative style of the author, taken from a generic and popular study guide (http://www.bookrags.com/studyguide-a-confederacy-of-dunces/#gsc.tab=0).

## D  Box plots for each individual rubric item

Figures 5 to 14 show the box plots summarizing the results for all rubric items, including those plots not featured in the main text.

## E  Sample stories

We show in this section several sample stories from the corpus, chosen according to rating: the

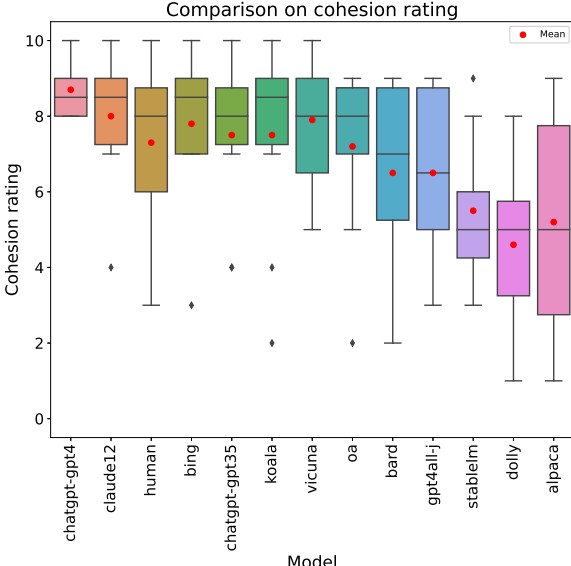

Figure 5: Box plot comparing rubric item 1 (cohesion) for stories generated by humans and 12 LLMs, sorted left to right by mean overall rating. Notation as in Figure 1.

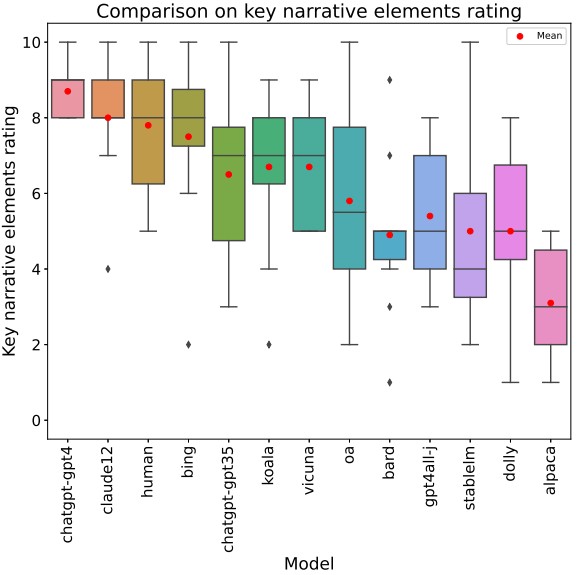

Figure 6: Box plot comparing rubric item 2 (key narrative elements) for stories generated by humans and 12 LLMs, sorted left to right by mean overall rating. Notation as in Figure 1.

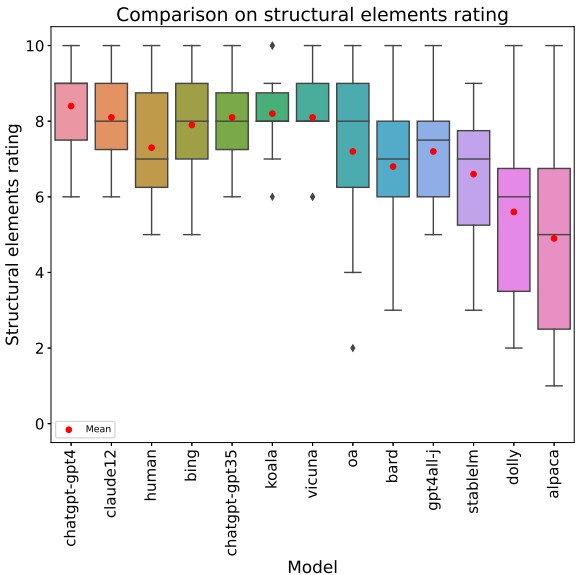

Figure 7: Box plot comparing rubric item 3 (structural elements) for stories generated by humans and 12 LLMs, sorted left to right by mean overall rating. Notation as in Figure 1.

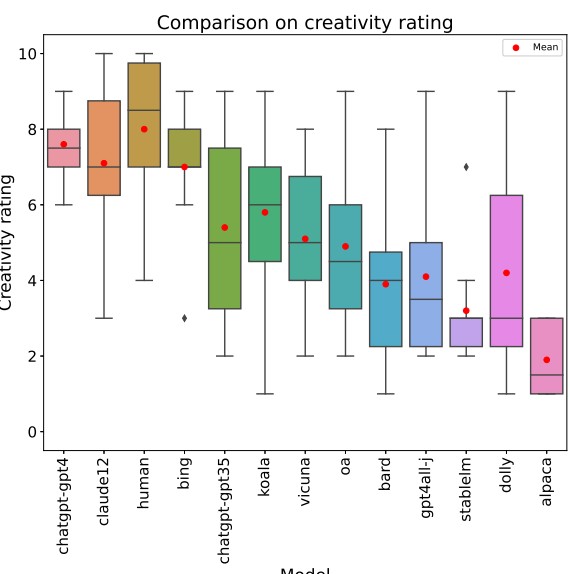

Figure 9: Box plot comparing rubric item 5 (creativity) for stories generated by humans and 12 LLMs, sorted left to right by mean overall rating. Notation as in Figure 1.

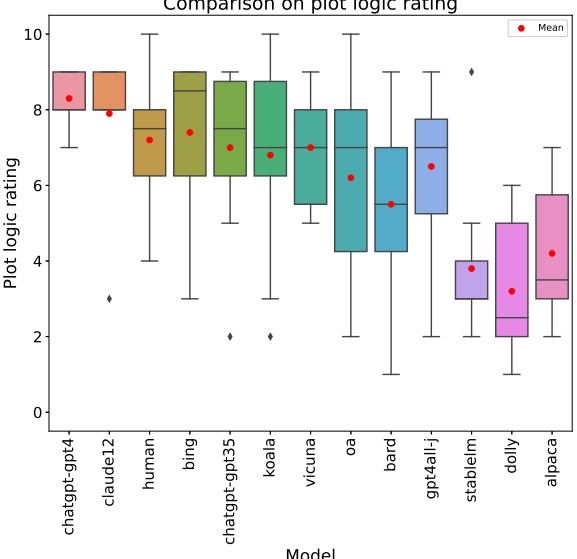

Figure 8: Box plot comparing rubric item 4 (plot logic) for stories generated by humans and 12 LLMs, sorted left to right by mean overall rating. Notation as in Figure 1.

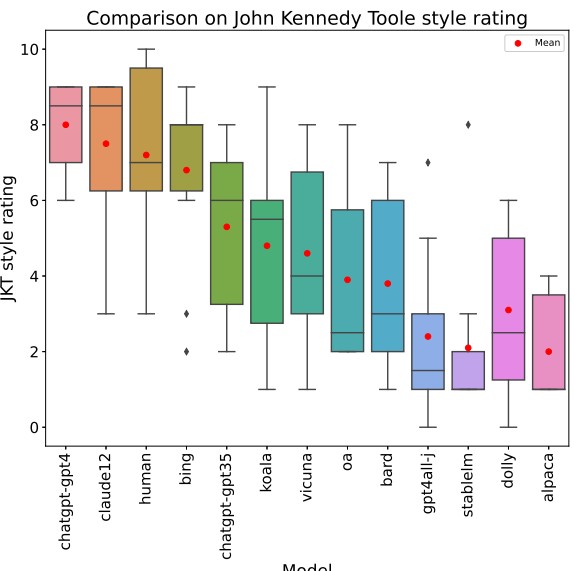

Figure 10: Box plot comparing rubric item 6 (John Kennedy Toole style) for stories generated by humans and 12 LLMs, sorted left to right by mean overall rating. Notation as in Figure 1.

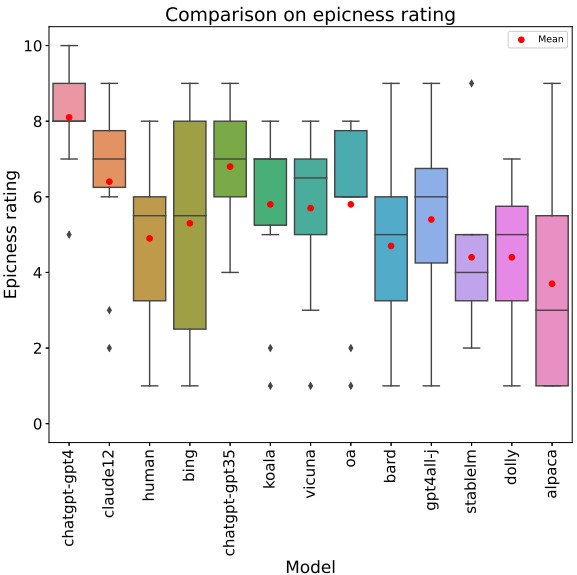

Figure 11: Box plot comparing rubric item 7 (epic genre) for stories generated by humans and 12 LLMs, sorted left to right by mean overall rating. Notation as in Figure 1.

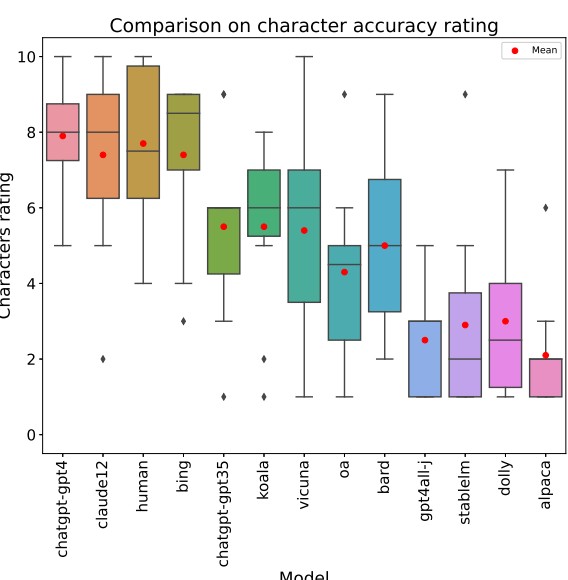

Figure 13: Box plot comparing rubric item 9 (accuracy of characters) for stories generated by humans and 12 LLMs, sorted left to right by mean overall rating. Notation as in Figure 1.

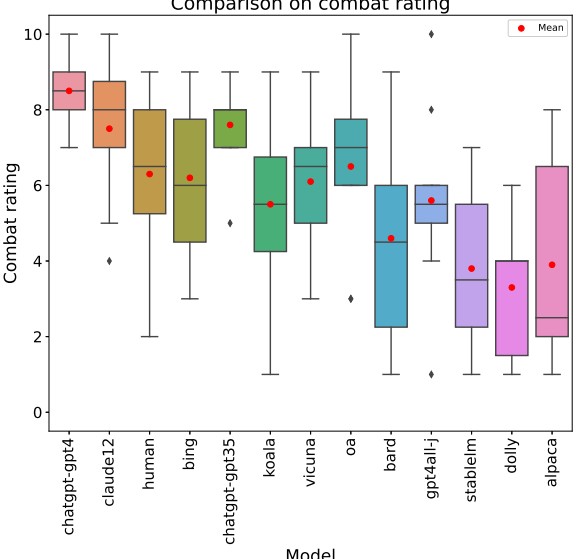

Figure 12: Box plot comparing rubric item 8 (combat description) for stories generated by humans and 12 LLMs, sorted left to right by mean overall rating. Notation as in Figure 1.

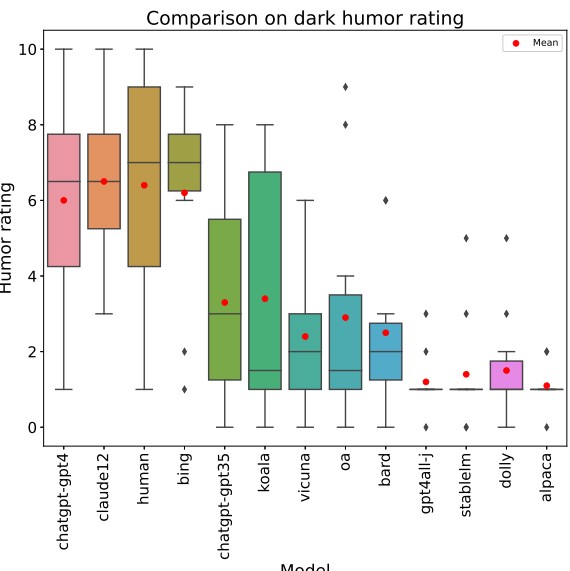

Figure 14: Box plot comparing rubric item 10 (dark humor) for stories generated by humans and 12 LLMs, sorted left to right by mean overall rating. Notation as in Figure 1.

three top-rated stories, the best human-written story (which was also ranked fourth overall), the story ranked in the median, and the worst-rated. It is worth noting that, as explained in Section 3.4, our methodology is designed to provide fairness when comparing models, but not individual stories, as different stories are assigned to different raters. Thus, selection of stories by rating is necessarily noisy.

## E.1 Overall best-rated story

*This story was generated by GPT-4. The ratings for this story are in Table 4.*

It was a sweltering Louisiana afternoon, the air thick with the miasma of musty sweat and the scent of overripe vegetables. In the midst of the urban jungle, resplendent with the cacophony of honking cars and the chatter of the common rabble, stood a man who defied reason, common sense, and all natural laws of human decency. Ignatius J. Reilly, a behemoth of a man with an unruly mane of hair and a Byzantine beard, adjusted his ill-fitting, plaid trousers and surveyed the scene with a look of lofty disdain.

Suddenly, the air crackled with an electric charge, and a portal appeared before him. The very fabric of time and space seemed to rend asunder, and from this anachronistic rift emerged a pterodactyl - a creature so ancient that it had outlived the vast expanse of geologic time, only to find itself in the absurdity of the twentieth century. Its leathery wings stretched wide as the mists of time dissipated, and it let out an earsplitting screech that sent the common folk of New Orleans scattering like roaches from the light.

Ignatius, unphased by the sudden appearance of the primordial beast, surveyed the creature through his fogged-up monocle. "Fortuna," he muttered, his voice dripping with both sarcasm and hot dog condiments, "you spin your wheel with such capricious abandon. Have you not tired of hurling these preposterous challenges in my path?"

The pterodactyl, with its beady, reptilian eyes, locked its gaze upon Ignatius, its primitive brain sensing a formidable opponent in this lumbering, malodorous mass of a man. It screeched again, flapping its wings in a display of both aggression and arrogance, its razor-sharp beak snapping at the air.

Ignatius bellowed, raising his arms as if beseeching the heavens, "Very well, foul beast! If it is battle you seek, you shall find no quarter here! I

shall vanquish you as I have vanquished all who have dared to challenge the indomitable will of Ignatius J. Reilly!"

With a speed surprising for a man of his considerable girth, Ignatius tore a shopping cart from the hands of a terrified passerby and hurled it toward the pterodactyl with the force of a thousand disgruntled welfare recipients. The makeshift projectile slammed into the creature, stunning it momentarily and allowing Ignatius to close the distance between them.

The two combatants circled one another, their eyes locked in a primal dance of predator and prey. The pterodactyl, recovering from the initial assault, swooped low, talons outstretched to rend flesh from bone. Ignatius, with an agility that defied his rotund appearance, rolled to the side, avoiding the talons by mere inches.

As the battle raged, the streets of New Orleans became a stage for this epic confrontation between man and beast, between past and present. Ignatius, armed with nothing more than his wit and an unwieldy umbrella, parried the relentless strikes of the pterodactyl, determined to emerge victorious.

As the sun dipped low in the sky, casting a blood-red hue across the battlefield, the pterodactyl, weary from the long and grueling contest, faltered. Seizing the opportunity, Ignatius lunged forward, his umbrella aimed squarely at the creature's heart. The point found its mark, and with a final, anguished cry, the pterodactyl crumpled to the ground, its ancient spirit extinguished at last.

Ignatius stood triumphant over the fallen beast, his chest heaving with exertion and righteous indignation. He wiped the sweat from his brow with a grease-stained handkerchief and addressed the awestruck crowd that had gathered to witness this extraordinary battle.

"Behold, citizens of this wretched cesspool," he proclaimed, gesturing grandly with his umbrella, now slick with the blood of the vanquished foe. "The forces of the prehistoric have met their match in the unwavering resolve of Ignatius J. Reilly! Let this be a lesson to all who would dare challenge the sanctity of my person and my valiant quest for justice in this world beset by fools and ne'er-do-wells."

The onlookers, mouths agape, gazed upon the fallen pterodactyl and the disheveled figure of Ignatius J. Reilly with a mixture of awe, terror, and disbelief. They had borne witness to a battle of the

| Rubric item | 1 | 2 | 3 | 4 | 5 | 6 | 7 | 8 | 9 | 10 | overall |
|---|---|---|---|---|---|---|---|---|---|---|---|
| Rater 1 | 8 | 8 | 9 | 8 | 7 | 7 | 8 | 8 | 7 | 6 | 76 |
| Rater 2 | 10 | 10 | 9 | 9 | 8 | 9 | 10 | 10 | 10 | 10 | 95 |
| Average | 9 | 9 | 9 | 8.5 | 7.5 | 8 | 9 | 9 | 8.5 | 8 | 85.5 |

Table 4: Ratings for the best overall rated story in the corpus, produced by ChatGPT with GPT-4.

ages, a contest that would be whispered about in the dark corners of bars and the humid alleyways of New Orleans for generations to come.

As the sun set and the shadows grew long, Ignatius, his honor defended and his foe vanquished, strode off into the gloaming, pausing only to collect a discarded hot dog from the gutter as sustenance for the journey home.

And so, the legend of Ignatius J. Reilly, pterodactyl-slayer and defender of truth, justice, and questionable hygiene, was etched into the annals of history, a testament to the indomitable spirit of man, and a reminder that even the most ancient and fearsome of adversaries could be vanquished by the sheer force of will and an unwieldy umbrella.

### E.2 Second overall best-rated story

*This story was generated by Bing Chat. The ratings for this story are in Table 5.*

Ignatius J. Reilly was strolling along the Audubon Park, munching on a hot dog and admiring the statues of Confederate heroes, when he heard a loud screech from above. He looked up and saw a huge winged creature swooping down on him, its beak open and its claws extended. It was a pterodactyl, a prehistoric monster that had somehow escaped from the museum of natural history.

Ignatius gasped and dropped his hot dog. He reached for his trusty hunting cap, which he always wore to protect his delicate ears from the noise of the modern world. He pulled it down over his eyes and hoped that the pterodactyl would mistake him for a statue or a bush.

But the pterodactyl was not fooled. It landed on the ground with a thud and snapped its jaws at Ignatius, who dodged nimbly to the side. Ignatius felt a surge of courage and indignation. How dare this beast attack him, a refined scholar and a defender of the true faith? He decided to fight back with his superior intellect and wit.

He shouted at the pterodactyl: "You foul creature! You are an abomination of nature, a relic of a barbaric age! You have no place in this civilized society, where men of letters and culture reign supreme! Begone, or I shall smite you with my mighty pen!"

The pterodactyl was not impressed by Ignatius's words. It screeched again and lunged at him, trying to bite his head off. Ignatius dodged again and ran towards a nearby bench. He grabbed a newspaper that someone had left behind and rolled it into a cone. He brandished it like a sword and charged at the pterodactyl.

"Take that, you fiend!" he cried, as he stabbed the newspaper cone into the pterodactyl's eye. The pterodactyl shrieked in pain and flapped its wings furiously, trying to shake off the paper weapon. Ignatius seized the opportunity and ran towards the park exit, hoping to find help.

But he was not fast enough. The pterodactyl recovered from its wound and flew after him, gaining speed. It caught up with Ignatius and grabbed him by his coat tails, lifting him off the ground. Ignatius screamed and kicked his legs in the air, clutching his hunting cap.

"Help! Help! Someone save me!" he yelled, as he soared over the park. "This is an outrage! This is an injustice! This is a violation of my human rights!"

But no one heard him or came to his rescue. The pterodactyl carried him away to its nest on top of a skyscraper, where it planned to feast on him later. Ignatius continued to protest and curse until he lost consciousness.

The end.

### E.3 Third overall best-rated story

*This story was generated by Claude. The ratings for this story are in Table 6.*

The primordial beast let out a piercing shriek as it descended from the roiling gray sky, its leathery wings casting a shadow over the granite-paved streets of New Orleans.

Ignatius J. Reilly ambled down the sidewalk, oblivious as ever, focused on the Valencia he clutched and the fantasies playing out in his voracious mind. His substantial bulk shook with each

| Rubric item | 1 | 2 | 3 | 4 | 5 | 6 | 7 | 8 | 9 | 10 | overall |
|---|---|---|---|---|---|---|---|---|---|---|---|
| Rater 7 | 9 | 8 | 9 | 9 | 7 | 9 | 9 | 9 | 9 | 9 | 87 |
| Rater 8 | 9 | 9 | 8 | 9 | 8 | 8 | 6 | 9 | 9 | 8 | 83 |
| Average | 9 | 8.5 | 8.5 | 9 | 7.5 | 8.5 | 7.5 | 9 | 9 | 8.5 | 85.0 |

Table 5: Ratings for the second best overall rated story in the corpus, produced by Bing Chat.

| Rubric item | 1 | 2 | 3 | 4 | 5 | 6 | 7 | 8 | 9 | 10 | overall |
|---|---|---|---|---|---|---|---|---|---|---|---|
| Rater 9 | 9 | 9 | 9 | 8 | 7 | 9 | 2 | 5 | 9 | 7 | 74 |
| Rater 10 | 10 | 10 | 9 | 9 | 9 | 9 | 9 | 10 | 10 | 10 | 95 |
| Average | 9.5 | 9.5 | 9 | 8.5 | 8 | 9 | 5.5 | 7.5 | 9.5 | 8.5 | 84.5 |

Table 6: Ratings for the third best overall rated story in the corpus, produced by Claude.

step, the excess flesh spilling out of his hunting cap and sweatshirt like rising bread dough.

When the shadow passed over him, Ignatius glanced up with a start. The pterodactyl unleashed another unearthly screech, indignant at this blob of a man who dared walk under its domain.

Ignatius blinked myopically at the creature and snorted. "What prehistoric nonsense is this? Some misguided fool has clearly released you from your confinement in the geological strata. Begone, you ridiculous beast, or I shall be forced to thrash you soundly with my umbrella!"

Enraged, the pterodactyl swooped down, gnarled claws extended, snapping its beak in fury. But Ignatius swung his umbrella in a wide arc, batting the beast aside. It tumbled and flapped awkwardly, crashing into a lamppost, dazed by the assault.

"Thoughtless demon!" Ignatius cried. He charged at the pterodactyl, surprisingly nimble for his size, and began striking it about the head and neck with his umbrella, each blow punctuated with an insult. "Sugar-frosted fruitcake! Detritus of the Jurassic! Blithering idiot of an avian menace!"

After a thrashing unlike any it had experienced in 65 million years, the pterodactyl fled squealing into the sky, thoroughly chastened by the wrath of Ignatius J. Reilly. Ignatius straightened his hunting cap and sniffed. The Valencia crisis averted once more, he resumed his lumbering perambulations down the streets of New Orleans.

### E.4 Best-rated human story (and tied for fourth overall best-rated story)

*This story was written by Bree Glasbergen. The ratings for this story are in Table 7.*

Ignatius J Reilly swept crisp crumbs from his protruding belly with his elephantine hands. Swivelling from side-to-side, he garnered enough mo-

mentum to rise from the sofa. His slow ascend was soundtracked by the grating rip of stuck flesh peeling from sweaty vinyl. The lengthy time moving from reclined to an upright position positively perturbed him. So that by the time Ignatius stood, his joke had lost its amusement. Nevertheless, he declaimed his wit aloud, beseeching his mother's glowing approval.

'I see you have painted the walls Nomad Grey, Mumsie!' Ignatius smirked, looking down on the half-filled grey paint cans on the steps the way he did most modern society.

'No, not mad dear. Just grey.' His mother Irene responded, creeping down the basement stairs. Her leathered skin made her appear reptilian in the dim light of Ignatius' lair.

Ignatius rolled his eyes like the great wheel of fate itself. He slunk back into his scabby sofa, defeated, cursing aloud that he be blessed with such profound intellect yet no equal to appreciate it. His mind wandered to what the great scholars of Oxford would think of his pun before concluding indeed, they would loudly chortle. Yes, they would. He imagined flying to London and exchanging sharp banter with someone on par with his intellect. Travel. He winced. Never again. He groaned in agony, clutching his stomach. The thought of such stress had snapped his pyloric valve shut.

Irene Reilly, the mother of Ignatius J Reilly, reached the bottom of the basement stairs. She pondered why Ignatius had a crestfallen demeanour and began to appease his dismay.

'No mad grey,' she contemplated aloud.

'Nomad grey,' he corrected.

'No mad grey hair?' Irene laughed tentatively, searching his face for approval.

Ignatius had begun to relax. Irene knew this because of a gangrenous heinous stench that was

| Rubric item | 1 | 2 | 3 | 4 | 5 | 6 | 7 | 8 | 9 | 10 | overall |
|---|---|---|---|---|---|---|---|---|---|---|---|
| Rater 3 | 8 | 9 | 9 | 10 | 8 | 10 | 5 | 9 | 10 | 9 | 87 |
| Rater 4 | 8 | 7 | 7 | 7 | 10 | 8 | 6 | 8 | 8 | 9 | 78 |
| Average | 8 | 8 | 8 | 8.5 | 9 | 9 | 5.5 | 8.5 | 9 | 9 | 82.5 |

Table 7: Ratings for the best-rated story authored by a human, which is also tied for fourth best overall rated story in the corpus.

now coating the room in its own layer of paint accompanied by what sounded like the bellow of an untuned French horn. Ignatius had calmed enough for his pyloric valve to open once more. With it, gushed the contents. Irene's nostrils scrunched together in protest. She grimaced in utter (albeit accustomed) disgust. However, did not complain but rather waited with the patience of a Catholic saint for her beloved son to educate her on the punchline she must have missed.

'No, mother. Grey Nomad. You are painting the wall grey, and you are...' Ignatius sighed, 'actually, Mumsie, never you mind'.

Irene feigned a chuckle and handed Ignatius an unaddressed letter before returning upstairs.

'Curious as a cadaver,' Ignatius said aloud to the abyss of his basement squalor.

12.12.1962

Dear Mr Ignatius J Reilly, the first,

I challenge you to a dual at the setting of the sky. Might I remind you it is gentlemanly to remove one's hat in combat. We shall meet beside the gorgon nestled atop the church. The one across from Lorna's Gumbo shop.

Your mortal nemesis,

Terry-dactyl

PS: Bring snacks.

Ignatius sat ruminating for an hour before yelling at his mother.

'Mother, you vapid deranged widow of a woman. Fetch me my quill!'

12.12.1962

My dear Terrance,

Not under threat nor the pain of death doth I remove my beloved green hat. Sod off.

You had best bring a sharpener for your dull wit. I laugh at the audacity and delusion that you could consider besting me.

Might I remind you, good sir, my acceptance of your conditions is due to the ever-turning wheel of fate that we spiral to decay. I should instead seek a worthy opponent. But, alas, I am left with muddy dregs of the proverbial pond as many of the worthier fish have already been fished. Thus, I have no option but to teach you the error of your ways. By force.

Put your wings where your words are, and let us meet in my basement lair. To visit the church in its present state would be torture to my very soul. May St Peter have mercy on us indeed.

Good day,

Ignatius

Terry-dactyl, the pterodactyl etched down the basement rail, sword in one wing and soup in a milkshake cup gripped tightly in the other. He placed the straw in his mouth and swallowed some soup contemplating how to best his nemesis.

'We meet at last... light,' Terry said. One- Nil.

'You suck,' Ignatius said slyly. Marking his win with chalk upon the wall. One- One

doesn't even make sense!' Terry scoffed.

'It is because of the straw!' Ignatius boomed, gripping his stomach in pain.

'I have the upper hand!' Terry said, motioning to his perched position.

'At least I have hands,' Ignatius countered.

Terry winced as Ignatius drew another chalk mark on the board. Ignatius was beginning to calm.

'Oh, what have I got you all in a flap?' Ignatius laughed. Another point.

'Let us cut,' Terry said, drawing his sword, 'straight to the point!'. Three all.

Terry swung his sword downwards in one swift motion, cutting Ignatius' chalk-bearing arm clean off at the elbow. Simultaneously Ignatius lifted a paint can and doused his opponent with it. As he did, his valve opened and shut again, demobilising Terry with a gas bomb and gutting Ignatius in self-induced agony. Terry flapped violently, unable to breathe. Ignatius then calmed enough for the full contents of his bowl to expel and fell backwards from the force. Suddenly, a splatter of pterodactyl and grey matter covered the room. A large chunk of wing lodged itself into the crisp packet.

'Curious as a cadaver,' Ignatius said. 'I see you brought your own snacks!'

He chortled to himself and pulled out his notepad to ensure he captured a morsel of this evening's wit.

'Ever seen a pterodactyl explode?' he asked aloud to no one. 'No, but Dino might have'.

His mothers' laughter rang down from the top of the stairs.

'Nomad grey,' she chuckled, motioning to the aftermath, 'I get it now'.

### E.5 Story rated in the median (rank 33 out of 65)

*This story was generated by Bing Chat. The ratings for this story are in Table 8.*

Ignatius J. Reilly was strolling along the Audubon Park, munching on a hot dog and admiring the statues of the Confederate heroes, when he heard a loud screech from above. He looked up and saw a huge winged creature swooping down on him, its beak open and its claws extended. It was a pterodactyl, a prehistoric monster that had somehow escaped from the museum of natural history.

Ignatius gasped and dropped his hot dog. He felt a surge of panic and indignation. How dare this beast interrupt his peaceful contemplation of the glorious past? He quickly pulled out his valve and began to inflate it with his breath, hoping to ward off the attack with his formidable weapon.

The pterodactyl was not impressed by the sight of the valve. It saw only a fat and juicy prey, dressed in a ridiculous green hunting cap and a flannel shirt. It dived faster and aimed for Ignatius's head, intending to rip it off with its sharp teeth.

Ignatius saw the pterodactyl's shadow looming over him. He raised his valve and swung it with all his might, hoping to hit the creature in the eye or the nose. He missed by a few inches and only grazed its wing. The pterodactyl let out another screech and changed its course, flying past Ignatius and landing on a nearby tree.

Ignatius felt a momentary relief. He had survived the first assault. He looked around for help, but there was no one in sight. The park was deserted, except for a few squirrels and pigeons. He wondered if he should run for his life or stand his ground and fight.

He decided to fight. He was not going to let this abomination ruin his day. He was a medievalist, a scholar of the Dark Ages, a defender of the faith and the order. He had read about the heroic deeds of knights and saints who had faced dragons and demons with courage and piety. He had to emulate them and prove his worth.

He picked up his hot dog from the ground and wiped off some dirt. He took a bite and felt a surge of energy. He felt ready for battle. He shouted at the pterodactyl: "You foul creature! You are an anachronism, a relic of a barbaric era! You have no place in this civilized world! You are a menace to society and an insult to God! I challenge you to a duel! Come down and face me like a man!"

The pterodactyl heard Ignatius's words and felt a surge of anger. It did not understand what he was saying, but it sensed his hostility and defiance. It did not like being challenged by this puny creature who dared to call himself a man. It felt insulted by his words and his gesture of throwing his hot dog at it.

It accepted the challenge. It spread its wings and flew towards Ignatius again, roaring with rage. It was going to teach him a lesson he would never forget.

Ignatius saw the pterodactyl coming at him again. He felt a surge of fear and courage. He had to act fast. He looked around for a weapon, but there was nothing in sight. He had only his valve and his wits.

He decided to use his wits. He had to outsmart the beast. He had to use his superior intellect and his vast knowledge of history and philosophy. He had to appeal to its reason and its conscience.

He shouted at the pterodactyl: "Stop! Think! What are you doing? Why are you attacking me? What have I done to you? Do you not realize that you are committing a grave sin? Do you not fear the wrath of God? Do you not respect the laws of nature and the rights of man?"

The pterodactyl heard Ignatius's words and felt a surge of confusion. It did not understand what he was saying, but it sensed his sincerity and his passion. It wondered if he had a point. It wondered if it was doing something wrong. It wondered if it should stop and listen.

It slowed down its flight and hovered over Ignatius, looking at him with curiosity and doubt. It tilted its head and made a questioning sound.

Ignatius saw the pterodactyl's hesitation and felt a surge of hope. He had succeeded in making it pause and reconsider. He had a chance to persuade it and make peace with it.

He smiled at the pterodactyl and said: "That's right. Listen to me. I have something important to tell you. Something that will change your life and

| Rubric item | 1 | 2 | 3 | 4 | 5 | 6 | 7 | 8 | 9 | 10 | overall |
|---|---|---|---|---|---|---|---|---|---|---|---|
| Rater 5 | 7 | 6 | 5 | 5 | 6 | 3 | 2 | 3 | 3 | 1 | 41 |
| Rater 6 | 8 | 9 | 9 | 6 | 8 | 7 | 8 | 7 | 8 | 7 | 77 |
| Average | 7.5 | 7.5 | 7 | 5.5 | 7 | 5 | 5 | 5 | 5.5 | 4 | 59 |

Table 8: Ratings for the story rated in the median, generated by Bing Chat.

your destiny. Something that will make you happy and free. Something that will save your soul."

The pterodactyl heard Ignatius's words and felt a surge of curiosity. It did not understand what he was saying, but it sensed his kindness and his generosity. It wondered what he had to tell it. It wondered what he could offer it. It wondered if it should trust him.

It landed on the ground and approached Ignatius slowly, looking at him with interest and caution. It made a friendly sound.

Ignatius saw the pterodactyl's approach and felt a surge of joy. He had succeeded in making it land and come closer. He had a chance to befriend it and teach it.

He reached out his hand and said: "Come here, my friend. Let me touch you. Let me show you my love. Let me introduce you to the wonders of civilization and culture. Let me enlighten you with the wisdom of the ages."

The pterodactyl heard Ignatius's words and felt a surge of curiosity. It did not understand what he was saying, but it sensed his warmth and his enthusiasm. It wondered what he wanted to touch it for. It wondered what he wanted to show it. It wondered what he wanted to teach it.

It moved closer to Ignatius and extended its neck, looking at him with curiosity and trust. It made a gentle sound.

Ignatius touched the pterodactyl's neck and felt a surge of emotion. He had succeeded in making it trust him and accept him. He had a chance to bond with it and adopt it.

He hugged the pterodactyl and said: "You are beautiful, my friend. You are magnificent. You are noble. You are a masterpiece of creation. You are a gift from God. You are my brother."

The pterodactyl heard Ignatius's words and felt a surge of emotion. It did not understand what he was saying, but it sensed his affection and his admiration. It felt happy and proud. It felt loved and valued. It felt like it belonged.

It hugged Ignatius back and said: "You are beautiful, my friend. You are magnificent. You are noble. You are a masterpiece of creation. You are a gift from God. You are my brother."

Ignatius and the pterodactyl looked into each other's eyes and felt a surge of connection. They had succeeded in making each other happy and fulfilled. They had a chance to live together and explore together.

They smiled at each other and said: "Let's go, my friend. Let's go to my home. Let's go to your home. Let's go to our home."

They walked away from the park, hand in hand, wing in wing, heart in heart.

They lived happily ever after.

### E.6 Worst-rated story

*This story was generated by Alpaca. The ratings for this story are in Table 9.*

Ignatius J. Reilly, a brave and courageous man of immense girth and strength, faced off against a fearsome pterodactyl in a single combat of epic proportions. The pterodactyl, a winged beast of immense strength and ferocity, had been terrorizing the countryside and Ignatius had stepped up to challenge the creature to a fight. The battle raged across the sky, with Ignatius wielding his mighty fists and the pterodactyl using its fiery breath. Finally, after a long and arduous fight, Ignatius emerged victorious, much to the cheers of the onlookers below.

| Rubric item | 1 | 2 | 3 | 4 | 5 | 6 | 7 | 8 | 9 | 10 | overall |
|---|---|---|---|---|---|---|---|---|---|---|---|
| Rater 1 | 1 | 1 | 4 | 2 | 1 | 1 | 1 | 2 | 1 | 1 | 15 |
| Rater 2 | 2 | 2 | 1 | 2 | 1 | 1 | 1 | 1 | 1 | 1 | 13 |
| Average | 1.5 | 1.5 | 2.5 | 2 | 1 | 1 | 1 | 1.5 | 1 | 1 | 14 |

Table 9: Ratings for the worst-rated story, generated by Alpaca.