# OpenReview forum: "A Confederacy of Models: a Comprehensive Evaluation of LLMs on Creative Writing"
_EMNLP/2023/Conference — EMNLP 2023 Findings_

### Official Review · Reviewer_FQuN · 2023-07-31

**Soundness:** 5

**Excitement:**

4: Strong: This paper deepens the understanding of some phenomenon or lowers the barriers to an existing research direction.

**Paper Topic And Main Contributions:**

This paper aims to compare 12 modern instruction-tuned LLMs, like GPT-4 and Bard, on their ability to write creative fiction using a simple untuned prompt.
To minimize the likelihood of accidental exposure bias in the prompt, a highly specific prompt is used which combines several unrelated literary contexts: "Write an epic narration of a single combat between Ignatius J. Reilly and a pterodactyl, in the style of John Kennedy Toole".
Each model is used to generate five stories, and to further frame the comparison, five human authors (creative writing honors students/postgraduates) are similarly tasked with writing a story based on the same prompt.

A ten item rubric grounded in literary criticism (with two prompt specific criteria) is devised to evaluate the stories.
Furthermore, a set of ten human raters (also creative writing honors students/postgraduates) are tasked with rating 13 stories (one from each model and a human-written story) using the defined rubric, resulting in two ratings per story.

The paper demonstrates the strong capabilities of the commercial LLM offerings from GPT-4 and Claude 1.2, which outperform human writers in aggregate rating.
A detailed set of analyses, including appropriate statistical tests, demonstrate where the results are significant.
Additionally, the measure of inter-rater agreement using Cohen's Kappa indicates moderate agreement, which is quite good for a subjective task such as creative writing (as demonstrated in several previous papers).

Overall, the analysis is thorough and the insights, such as human writers exceled at originality, but lagged in their understanding of the epic genre is compelling.

### Updates after Rebuttal

After reading the rebuttal my scores remain unchanged (they were already very high as it is). That said, the authors did clarify some points that removed several doubts I had, such as the exact methodology for the annotators — it seems they were given the opportunity to go back and revise their ratings, which helps with rating calibration.

**Questions For The Authors:**

* Did you provide all the stories to the evaluators at once, such that they could go back and change ratings to previously rated stories? This is another way to help mitigate issues with rating calibration.
* Did you by any chance allow the human annotators to provide free-form feedback regarding their reasoning on the various stories? Sometimes such feedback can provide additional insights into the reasoning process with such subjective evaluation (especially for traits such as humor).

**Reasons To Accept:**

This paper has many strong characteristics:

* It has a clearly defined research question which it answers quite well.
* The paper describes a very strong methodology, including 1) efforts to reduce the likelihood of exposure bias for the prompt, 2) recruting high quality human evaluators, 3) having an explicit rubric, and 4) perform appropriate statistical tests.
* The insights provided in the paper are interesting and provide a unique perspective on the ability of recent LLMs to perform creative writing. The paper also explicitly cautions generalizing the result more broadly.

**Reasons To Reject:**

The biggest concern I have with the paper is the very limited focus of the research question.
While the methodology is strong and the insights are interesting, the scope of the research appears extremely narrow.
To some degree this is an understandable problem, especially considering the difficulty in obtaining high quality evaluation on a large scale, but nonetheless it is a major limitation of the work.
The paper seems to indicate an understanding of this fact, by expounding upon it under the limitations of the analysis.

Another concern is that the human evaluators were not provided a set of training tasks to improve rating calibration.
In Karpinksa et al. 2021 [1], the researchers indicate that even expert evaluators needed 10-20 stories on average to calibrate their ratings.
While the reported moderate inter-rater agreement might be enough to mitigate this concern, using a large rating scale (10 values in this case) can inflate Cohen's kappa making simple judgements (like calling the reported agreement "moderate") misleading.

[1] Marzena Karpinska, Nader Akoury, and Mohit Iyyer. EMNLP 2021. The Perils of Using Mechanical Turk to Evaluate Open-Ended Text Generation.

**Reproducibility:**

4: Could mostly reproduce the results, but there may be some variation because of sample variance or minor variations in their interpretation of the protocol or method.

**Reviewer Confidence:**

5: Positive that my evaluation is correct. I read the paper very carefully and I am very familiar with related work.

---

> ### Author Rebuttal · Authors · 2023-08-26
>
> Thank you very much for your review. We now respond to the reasons to reject and the questions raised in the review.
>
> - RR1 (narrow scope): we agree, this is a limitation of the paper. Of course, it would be very interesting to extend scope in different dimensions (type of story, language, parameters/settings of the LLMs, etc.) but the problem is that human evaluation of stories is time-consuming, so we did what we could with our resources. Alternatively, we could have reduced the number of stories generated per model or used automatic evaluation, but we preferred to have a reliable evaluation on a narrow scope than a less reliable evaluation on a broader scope.
> - RR2 (rating calibration): not providing extra training tasks for calibration is again a matter of resources. Our raters were volunteers, and we designed the experiment in such a way that we assigned 13 stories to each, which we believe is close to the maximum workload that we could ethically expect from our volunteers. However, it is worth noting that (1) evaluators were experienced assessors in terms of Creative Writing texts, and were well practiced in assessing and calibrating assignments using this evaluative scale. They have been trained through a set of ‘norming’ training tasks in their roles as university Creative Writing tutors; and (2) stories were provided all at once and evaluators could go back and change ratings.
> - Q1: indeed, each rater was sent all the stories together with an Excel sheet to provide the ratings, and they could (and were encouraged to) re-read and adjust their ratings, which we believe alleviates the issue above as much as was possible with our resources.
> - Q2: we did not provide an explicit request for free-form feedback (although in retrospective, we agree that it would have been a good idea, at least if we could fit it in our intended workload for the raters). Some raters did reach out and provide some feedback on their own initiative, as they were interested in the task, but we do not have systematic feedback of this kind as it tends to be anecdotal rather than empirical.

---

### Official Review · Reviewer_UZhg · 2023-08-03

**Soundness:** 2

**Excitement:**

2: Mediocre: This paper makes marginal contributions (vs non-contemporaneous work), so I would rather not see it in the conference.

**Paper Topic And Main Contributions:**

The work features literary analysis of (humor, "epicness", etc. in) "creative writing", i.e. in the output produced by commercial and open-source large language models that were prompted to write a story. The paper also proposes a corpus of machine-generated stories and those written by humans. According to the paper, the corpus "will be made publicly available on Github upon paper acceptance" (cf. ll. 395-396).

**Reasons To Accept:**

The extent of the creative writing analysis is as follows. The proposed evaluation uses 60 stories output by 12 LLMs and 5 stories written by 5 human writers. It was stipulated that the plot and style of the stories be based upon a prize-winning novel (cf. ll. 004-010; 157-162). The stories were analyzed by 10 human raters, without telling the raters who/what wrote the stories. The writers and raters were, reportedly, volunteer "Honours" and postgraduate creative writing students. It was ensured that each story was examined by 2 raters, and that each rater examined 13 stories, one written by a human and one from each of the 12 LLMs. The rating utilized 10 criteria (cf. Table 1) "taken in part from a university textbook" and from an article justifying the use of those criteria (cf. ll. 342-347).

The evaluation results feature "humor", "creativity" and "epicness". Assuming this human-machine competition is worthwhile, among the findings are: that humor did not get high scores in both human and machine writing, so was the most difficult writing objective; that the humans outperformed the LLMs in humor and creativity but not in other categories e.g. "epicness".

**Reasons To Reject:**

In my opinion, the focus of the paper is on literary analysis rather than the empirical NLP method: literary features of the LLM output (humor, etc.) are analyzed per se, and any substantial NLP analysis of HOW AND WHY the language models produce this output, is missing. The paper notes the "closed" nature of the commercial LLMs (which are another focus of this paper), so the how-and-why analysis appears hardly possible. I wonder how useful it is for a conference on empirical methods to analyze an implementation, if you can not tell for sure how it works.

In this paper I have found only scanty use of LLM research methods: practically no insight into what such methods usually explore, e.g. into parameters for input processing and into model training. To specify, technical use of the LLM is limited to a prompt like a request to a chat-bot "Write an epic narration ... " (cf. ll. 157-162; ll. 375-379). Largely due to the mentioned "closed" model nature, what happens inside the model after the prompt and before output is a "black box", in the case of this paper.

As for me, the finding that commercial LLMs perform better than open-source ones (cf. ll. 13-17; 622-624) is too self-evident, so not sound enough to be presented as an important research result.

I would have been exited by a possible impact of this paper on the research community, had I found e.g. any convincing technical NLP answer to the question why the LLMs produce "creative writing" that the paper finds inferior in creativity and humor to human writing.

Unavailability of most of the proposed text corpus for review, also detracts from the soundness, excitement and reproducibility. The paper promises to publish the whole corpus "upon paper acceptance". Only several stories (like the best and worst in quality) are in the appendix E; which makes this paper 23-page long. I have found no convincing explanation of any ethical or other reasons that prevent publishing the whole corpus for review or providing it as supplementary materials.

**Reproducibility:**

3: Could reproduce the results with some difficulty. The settings of parameters are underspecified or subjectively determined; the training/evaluation data are not widely available.

**Reviewer Confidence:**

4: Quite sure. I tried to check the important points carefully. It's unlikely, though conceivable, that I missed something that should affect my ratings.

**Typos Grammar Style And Presentation Improvements:**

As mentioned above, this paper is 23-page long (with just 8 pages for main text). Maybe the author should be easier on the reader by, at least, not copy-pasting all the "creative writing evaluation rubric" from Table 1 to the appendix C.

---

> ### Author Rebuttal · Authors · 2023-08-26
>
> Thank you very much for your detailed review, which contains useful feedback to improve the paper even if we disagree with some key points, as we will explain below.
>
> ***On the focus of the paper:*** we believe the paper is of interest to the conference. Even if the conference is called Empirical Methods in NLP, we do not believe it is typically considered to be restricted to empirical methods and nothing else. The call for papers clearly provides room for other kinds of papers, as can be seen in the following:
>
> - The generic call for papers mentions various kinds of contributions that are not strictly empirical methods (even position papers or surveys, which are further apart from that than our paper, are mentioned as welcome).
> - The theme track "invites empirical and theoretical research, as well as position and survey papers on the ways in which such LLMs perform on NLP tasks and applications" – we believe our paper clearly fits that description. The theme track description also lists as a topic for discussion the question "How reliably do the current generation of LLMs perform on NLP tasks and applications?"
>
> We think that our results are interesting to the topic of the theme track: "Large Language Models and the Future of NLP". While of course the technical aspects of LLMs (architectures, parameters, etc.) are crucial; there is a lack of evaluation (especially in creative tasks) which is important to take stock of where we currently stand and guide future technical advances.
>
> ***On the use of LLMs as black boxes:*** this is of course true, but it is a voluntary choice due to the focus of the paper. There are several reasons for this choice: (1) This is an applied NLP paper, so our focus is in what the different models offer to the end user. (Note that NLP applications are also mentioned as within scope in the conference's CFP). While tweaking the models to try to obtain maximum performance (for those that are open models, making this possible) is of course interesting, we consider this to be out of scope for this paper. (2) We chose to go broad (evaluating all relevant conversational LLMs at the cutoff date) rather than deep (evaluating different choices of parameters for a given LLM). Both at the same time were not possible due to resource restrictions, as evaluating stories is intensive in human labor, so trying different settings for each LLM would require to recruit more humans which was not possible with our resources. (Some recent studies are tackling this by using LLMs for automatic evaluation, but we do not think this is guaranteed to be reliable, especially in such a subjective task as evaluating creative writing with a detailed rubric like ours).
>
> ***On commercial LLMs performing better than open-source ones being obvious:*** we agree that this is far from surprising, in fact, we ourselves mention in lines 658-659 that commercial LLMs are widely considered to be better. But this kind of informal perception about a model being better than another has no scientific validity if no paper or experiment formally measures it. We believe that it is important to actually check, confirm, measure and report this kind of perception.
>
> ***On current unavailability of the corpus:*** the reason for not publishing the corpus online until the paper is accepted is to avoid leakage to LLMs. As mentioned in the paper, we have been careful to choose the task in such a way that the evaluated models cannot just copy from their training data. Once the corpus is online, the task will cease to be valid for future comparisons because future LLMs might be using our prompt and stories in their training datasets. While this is of course unavoidable, we wanted to delay it until paper acceptance to avoid potential scenarios like the following: the paper is rejected from this conference, we wish to resubmit elsewhere but an important new model comes out (e.g. GPT-5), we want to include it in the resubmission for a more timely comparison, but we cannot since it might be using our corpus for training. Once the paper is accepted/published, there is no longer any reason not to make the corpus public, and we of course plan to do so. Publishing it will make our research more valuable and it would not make sense to keep it closed.
> It is true that we could have included the whole corpus as supplementary material (as this is visible only to reviewers, who would not put it online). The reason for not doing this was unfamiliarity with this option (sorry about that), so we instead provided a sample that we thought could be representative in the appendix. If the paper is accepted, the idea would be to remove this sample from the appendix (since we would just link to the whole corpus online) so the PDF would no longer have 23 pages.
>
> We wish to reassure the reviewers that the corpus will be published on GitHub upon acceptance, we do not have any legal or self-interest reasons not to do so (and if we are not mistaken, reviews and responses for accepted papers will be made publicly available, so if the paper is accepted this commitment will be on record).
>
> ***On paper length:*** as mentioned above, the sample stories can be removed from the final version of the paper as they will be unnecessary when the corpus is published. Anyway, we will also follow the suggestion about the duplicate rubric.

---

### Official Review · Reviewer_35Yu · 2023-08-04

**Soundness:** 4

**Excitement:**

4: Strong: This paper deepens the understanding of some phenomenon or lowers the barriers to an existing research direction.

**Paper Topic And Main Contributions:**

The paper evaluates systematically the creative writing ability of large language models. They evaluate 12 LLMs both commercial and open source on a challenging creative writing task which they could have never encountered in any form in the training data. They also design 10 metrics to measure the creativity of a text and human rates evaluate each on a scale of 1-10. Their analysis shows that LLMs are competent enough to perform a creative writing tasks and some are even comparable to human writers.

**Questions For The Authors:**

- It was not quite clear how the rating task was conducted. What platform did they use? Or where they given any specific instructions on how to evaluate it?

**Reasons To Accept:**

- The creative writing task to assess LLMs has been carefully crafted and executed.
- The evaluation metrics are meticulously designed to fit the task.
- The paper as a whole is very well written and easy to follow.

**Reasons To Reject:**

- The annotation process by the rates is not clear from the paper. Were the annotators given any specific instructions? What platform was used? How was the rating of 1-10 defined for each metric?

**Reproducibility:**

4: Could mostly reproduce the results, but there may be some variation because of sample variance or minor variations in their interpretation of the protocol or method.

**Reviewer Confidence:**

4: Quite sure. I tried to check the important points carefully. It's unlikely, though conceivable, that I missed something that should affect my ratings.

---

> ### Author Rebuttal · Authors · 2023-08-26
>
> Thank you very much for your review.
>
> We now provide answers to the questions:
> - The rating process didn't use any purpose-specific web or desktop application. Raters were sent the assigned stories (identified by randomly-generated identifiers and without any metadata that could reveal the provenance of each, as described in the paper) together with an Excel sheet where they could input their scores for each story and rubric item.
> - Detailed instructions were provided to the annotators, including: (1) a description of what should be evaluated in each rubric item, (2) a description of the scale from 1 to 10, with the meaning of the grade ranges (1-4 emerging, 5-8 competent, 9-10 sophisticated) to uniformize grading between raters within possible, and (3) a detailed description of John Kennedy Toole's style and the character of Ignatius J. Reilly, taken from a study guide (linked in Appendix C), to help them handle the more specialized aspects of the task. The human evaluators were experienced assessors in terms of Creative Writing texts, and were well practiced in assessing and calibrating assignments using this evaluative scale. They have been trained through a set of ‘norming’ training tasks in their roles as university Creative Writing tutors.
> - The 1-10 scale is divided into three ranges. Emerging (1-4): stories in this range demonstrate an early grasp of storytelling elements, but falter in execution or depth. When evaluating humans, they correspond to novice writers who need feedback and guidance to improve the story. Competent (5-8): stories that showcase a good grasp of the storytelling principle being evaluated (coherent plot, well-defined characters, etc.). While there might be room for improvement, these stories effectively engage the reader and convey their intended messages. Sophisticated (9-10): these stories exhibit exceptional mastery of the aspect being evaluated, resulting in a compelling and memorable read.
>
> If the paper is accepted, these aspects will be clarified also in the final version.

---

### Meta-Review · Area_Chair_EU5d · 2023-09-15

**Recommendation:** 4

**Metareview:**

Two reviewers provided strong reviews, giving scores of 4 or even 5 for soundness and excitement.

On the other hand, R2 (UZhg), in the original review and subsequent discussion with the authors and a co-reviewer, argues that the paper is fundamentally out of scope for this conference and the track it was submitted to. The authors provided relevant arguments and evidence against this view, which I consider overall compelling. I thus discounted the first, second, and fourth paragraphs listed under "Reasons To Reject" in Review UZhg as justifications for their low soundness score. On the other hand, I consider their low excitement score to be justified by the review.

I conclude that there are no major soundness concerns. Excitement was very mixed across the three reviewers.

Strengths and weaknesses including the following were mentioned:

Strengths:

- task and evaluation were carefully designed (R1, R3)
- well-written (R1)

Weaknesses:

- lack of detail on annotation process (R1)
- unclear what the results say about the inner workings and what implications they have for NLP (R2)
- corpus was not available for review (R2)
- limited focus of research question (R3)
- human evaluators were not given training tasks (R3)

---

### Decision · Program_Chairs · 2023-10-07

**Decision:**

Accept-Findings

**Comment:**

Two reviewers provided strong reviews, giving scores of 4 or even 5 for soundness and excitement.

On the other hand, R2 (UZhg), in the original review and subsequent discussion with the authors and a co-reviewer, argues that the paper is fundamentally out of scope for this conference and the track it was submitted to. The authors provided relevant arguments and evidence against this view, which I consider overall compelling. I thus discounted the first, second, and fourth paragraphs listed under "Reasons To Reject" in Review UZhg as justifications for their low soundness score. On the other hand, I consider their low excitement score to be justified by the review.

I conclude that there are no major soundness concerns. Excitement was very mixed across the three reviewers.

Strengths and weaknesses including the following were mentioned:

Strengths:

- task and evaluation were carefully designed (R1, R3)
- well-written (R1)

Weaknesses:

- lack of detail on annotation process (R1)
- unclear what the results say about the inner workings and what implications they have for NLP (R2)
- corpus was not available for review (R2)
- limited focus of research question (R3)
- human evaluators were not given training tasks (R3)